# Design and Control of a Piezoelectric-Driven Microgripper Perceiving Displacement and Gripping Force

**DOI:** 10.3390/mi11020121

**Published:** 2020-01-21

**Authors:** Yanru Zhao, Xiaojie Huang, Yong Liu, Geng Wang, Kunpeng Hong

**Affiliations:** 1The College of Mechanical and Power Engineering, Henan Polytechnic University, Jiaozuo 454003, China; 211705010024@home.hpu.edu.cn (X.H.); wgmouse@163.com (G.W.); saveit@163.com (K.H.); 2Exquisite Automotive Systems Company Limited Exquisite Stamping and Welding Branch Company, Baoding 071000, China

**Keywords:** microassembly, microgripper, PID controller, tip displacement, gripping force

## Abstract

A piezoelectric-driven microgripper with three-stage amplification was designed, which is able to perceive the tip displacement and gripping force. The key structure parameters of the microgripper were determined by finite element optimization and its theoretical amplification ratio was derived. The tracking experiments of the tip displacement and gripping force were conducted with a PID controller. It is shown that the standard deviation of tracking error of the tip displacement is less than 0.2 μm and the gripping force is 0.35 mN under a closed-loop control. It would provide some references for realizing high-precision microassembly tasks with the designed microgripper which can control the displacement and gripping force accurately.

## 1. Introduction

With the rapid development of micro-electromechanical systems (MEMS), microassembly and micromanipulation as the key technologies have become a hot research topic at home and abroad gradually [1]. The microgripper is an end effector for precision micromanipulation systems [2,3,4], the performance of which will affect the quality and accuracy of micro-operation process directly [5,6]. Therefore, it is of great practical significance to research on efficient and reliable microgripper.

According to different driving modes, microgrippers can be divided into six major types: The electrostatic-driven microgrippers [7,8], electrothermal-driven microgrippers [9], electromagnetic microgrippers [10], piezoelectric-driven microgrippers [11,12,13], shape memory alloy microgrippers [14,15] and vacuum microgrippers [16]. The piezoelectric stack actuators are used in various high precision microgrippers due to the advantages of fast response, high resolution, and large driving force [17,18,19].

The output displacements of piezoelectric stack actuators are usually ten or more micrometers, however the sizes of objects manipulated are generally several hundred micrometers, so a displacement amplifying mechanism is required to amplify and transmit the displacement from the actuator to the gripping jaws. Based on compliant mechanism, Cui et al. [20] designed a two-stage amplification microgripper and tested the tip displacement and gripping force. Yang et al. [21] devised a piezo-driven microgripper with bridge-type mechanisms and parallelogram mechanisms, in which the gripping range is 0.345–328.2 µm, the amplification ratio is 16.4, and the natural frequency is 157.5 Hz. In addition, by applying right circular leverage mechanisms, Wang et al. [1] proposed and analyzed a compliant microgripper. The maximum output displacement is 150 µm and the corresponding amplification ratio is 16.

In many micro-operation tasks, due to the irregular shape of some objects manipulated, the gripper tips and the micro-objects have a tendency to slip, which affects the efficiency and precision of the micro-operation process, and makes the control of the microgripper more complicated. Parallel grasping can avoid the problem effectively [19]. In addition, it is necessary to regulate both the tip displacement and the gripping force precisely because the objects manipulated are small and can be readily damaged. The resistance strain sensors were used to detect and control the tip displacement and the gripping force due to the properties of high resolution, simple mechanism, low price, and mature technology.

## 2. Structural Model Setting-Up and Finite Element Analysis

### 2.1. Structural Model Setting-Up and Analysis

The mechanical structure of the piezoelectric-driven microgripper designed consists of a three-stage amplification mechanism, piezoelectric stack actuator (PSA), tip displacement sensor, force sensor, gripping jaw, preload bolt, and gasket, as shown in Figure 1. A_1_, A_2_, A_3_, A_4_, C, D_1_, D_2_ are single-notch flexible hinges, E, F are double-notch flexible hinges, and B is a rectangular flexible hinge. In order to get a large tip displacement, a three-stage amplification mechanism including a triangular amplification structure and two leverages amplification structures were designed. In view of the symmetric characteristic, the left half of the kinematic diagram of mechanism of the microgripper is depicted in Figure 2. The motion principle of microgripper is: Under the actuation of piezoelectric stack actuator, the link D_1_F rotates clockwise around hinge F and drives link D_1_C to make hinge C move clockwise around D_2_; link BC moves to the right and drives link A_2_A_3_ to rotate clockwise around hinge A_3_ through hinge B; since A_1_–A_4_ is a parallelogram mechanism, the translational motion to the right of link A_1_A_2_ can drive the gripping jaws.

The aluminum alloy was adopted as the material of microgripper by reason of well comprehensive mechanical properties, namely low density, high elasticity, good toughness, and moderate strength. With finite element optimization, the relevant parameters of the microgripper were determined, as listed in Table 1. In Table 1, r_A_, r_C_, r_D_, r_E_, r_F_ represents the radii of the flexible hinges A, C, D, E, F, respectively and t_A_, t_C_, t_D_, t_E_, t_F_ the thicknesses; d is the width of the rectangular flexible hinge B; l_1_–l_6_ are the lengths of each part in Figure 2; γ is the angle between link D_1_C and the y-axis negative direction; h is the thickness of the gripping jaw, and b is the thickness of the microgripper.

The three-stage amplification mechanism is built. Figure 3 shows the pseudorigid-body model of the three-stage amplification mechanism.

Based on the geometric and motion relationships, the following formulas can be obtained:
(1)$$l_{AB}e^{i\varphi_{1}}+l_{BC}e^{i\varphi_{2}}=l_{CG}e^{i\varphi_{5}}+l_{AG}e^{i\beta }$$
(2)$$l_{AB}e^{i\varphi_{1}}+l_{BE}e^{i\varphi_{3}}=l_{AD}e^{i\alpha }+l_{DE}e^{i\varphi_{4}}$$
where α and β are the angles between links AD, AG, and the x-axis positive direction, *l_AB_*, *l_BC_*, *l_BE_*, *l_AD_*, *l_DE_*, *l_CG_*, *l_AG_* are lengths of links AB, BC, BE, DE, CG, AG, respectively, *φ*_1_, *φ*_2_, *φ*_3_, *φ*_4_, *φ*_5_ are the initial angular positions of links AB, BC, BE , DE, CG, respectively.

Formulas (2) and (3) can be rewritten based on Euler’s formula, namely:
(3)$$l_{AB}\cos\varphi_{1}+l_{BC}\cos\varphi_{2}+i(l_{AB}\sin\varphi_{1}+l_{BC}\sin\varphi_{2})=l_{CG}\cos\varphi_{5}+l_{AG}\cos\beta +i(l_{CG}\sin\varphi_{5}+l_{AG}\sin\beta )$$
(4)$$l_{AB}\cos\varphi_{1}+l_{BE}\cos\varphi_{3}+i(l_{AB}\sin\varphi_{1}+l_{BE}\sin\varphi_{3})=l_{AD}\cos\alpha +l_{DE}\cos\varphi_{4}+i(l_{AD}\sin\alpha +l_{DE}\sin\varphi_{4})$$


Differentiate Formulas (3) and (4) with respect to time and let the real and imaginary parts be equal. The following formula can be obtained:
(5)$$A\;\omega =B\;\omega_{5}$$
where $A=\left[\begin{array}{cccc} l_{AB}\sin\varphi_{1} & l_{BC}\sin\varphi_{2} & 0 & 0\\ l_{AB}\cos\varphi_{1} & l_{BC}\cos\varphi_{2} & 0 & 0\\ l_{AB}\sin\varphi_{1} & 0 & l_{BE}\sin\varphi_{3} & -l_{DE}\sin\varphi_{4}\\ l_{AB}\cos\varphi_{1} & 0 & l_{BE}\cos\varphi_{3} & l_{DE}\cos\varphi_{4} \end{array}\right]$,

$\omega =[\begin{array}{c} \omega_{1}\\ \omega_{2}\\ \omega_{3}\\ \omega_{4} \end{array}]$, $B=[\begin{array}{c} l_{CG}\sin\varphi_{5}\\ l_{CG}\cos\varphi_{5}\\ 0\\ 0 \end{array}]$.

Where *w*_1_*–w*_5_ are the initial angular velocity of links AB, BC, BE, DE, CG, respectively.

The amplification ratio of the microgripper can be computed as follows:
(6)$$\lambda =\frac{d_{out}}{d_{in}}\approx \frac{\partial_{d_{out}}}{\partial_{d_{in}}}=\frac{\omega_{4}l_{DF}}{\omega_{5}l_{in}}=A^{-1}B\left(4,1\right)\frac{l_{DF}}{l_{in}}=\frac{l_{CG}l_{DF}\sin\left(\varphi_{2}+\varphi_{3}\right)\sin\left(\varphi_{2}-\varphi_{5}\right)}{2l_{DE}l_{in}\sin\varphi_{2}\cos\varphi_{2}\sin\left(\varphi_{3}-\varphi_{4}\right)}$$
where *l_DF_* is the length of link DF, *l_in_* is the distance between the input end and hinge G.

### 2.2. Finite Element Analysis of the Microgripper

In order to verify the theoretical amplification ratio of the microgripper, finite element analysis (FEA) was performed using the ANSYS Workbench. The material properties of the microgripper are shown in Table 2.

A 10 μm input displacement was imposed on the microgripper and the displacement cloud diagram and stress cloud diagram are shown in Figure 4. It was shown in Figure 4a that the maximal output displacement of the gripping jaw is 191.25 μm, that is, the amplification ratio of the microgripper is about 19.1. The theoretical result of the amplification ratio calculated with Formula (6) is 23.5, namely, the relative error is 23%. The deviation arose because the models used for theoretical calculation were idealized. It was found from Figure 4b that the stress of the microgripper was concentrated on the flexible hinge, in which the maximal stress on the flexible hinge D_1_ was 90.7 MPa, which was much smaller than the yield limit of the material. Therefore, there is no yield failure in actual use. 

In order to verify the translational gripping effect of the microgripper, a 10 μm input displacement was imposed on the microgripper. The output displacement of the left gripping jaw of the microgripper increases gradually from the tip to bottom. The maximum output displacement was 191.25 μm at the bottom of the gripping jaw and the minimum 190.82 μm at the tip. The translational error produced by the left gripping jaw is only 0.43 μm, which indicated that the microgripper has fine translational performance, and can achieve the translational gripping of the operating object. Figure 5 shows the displacement cloud diagram of the left gripping jaw of the microgripper. 

## 3. Results and Discussion

### 3.1. Experiment Platform

In order to test the performance of the microgripper, an experiment system including the microgripper, driving power, data acquisition card (PCI-6221 from NI Corp, Shanghai, China with D/A and A/D modules), capacitive sensor, piezoelectric stack actuator (AE203D08F from Thorlabs Corp, Shanghai, China), strain signal conditioning circuit, strain gauges (BFH120-3AA-D100 from Xintai Corp, Shenzhen, China, the resistance 120 ohms and sensitivity coefficient 2.0% ± 1%) and the computer was established. The working process is as follows: A digital signal with certain waveform was generated by the computer with MATLAB software, and then was converted into an analog signal by the D/A module on the data acquisition card; the analog signal was amplified to 0–100 V by the driving power, and then to actuate the piezoelectric stack actuator; the voltage signal detected by the strain gauges integrated on the body of the microgripper was amplified with the strain signal conditioning circuit and then transmitted to the low-pass filter created by Simulink in the computer for filtering through the data acquisition card. The experimental platform is shown in Figure 6.

### 3.2. The Test of Tip Displacement and Gripping Force of the Microgripper

The piezoelectric stack actuator was applied a sinusoidal driving voltage with the frequency of 0.1 Hz and amplitude of 100 V, and the output displacement and gripping force of the microgripper can be obtained, as shown in Figure 7. We can see that the maximal output displacement of the microgripper was approximately 102.3 μm, and the corresponding input displacement was 6.1 μm, so the amplification ratio of the microgripper can be derived, as 16.8. It can also be found from Figure 7b that the maximal gripping force of the microgripper was 227.7 mN.

### 3.3. PID Feedback Control of the Microgripper

In order to improve the accuracy of displacement and force for the microgripper, the PID controller was used. The controlling discipline of the PID controller is shown in the following Formula (7). The PID controller is divided into there parts: The proportional part, integral part, and differential part. The functions of the three parts are as follows: The proportion regulator can speed up the response speed and improve the regulation accuracy of the system; the integration regulator is conducive to eliminate the steady-state error and improve the control accuracy of the system; the differential regulator can improve the dynamic performance of the system effectively.
(7)$$u\left(t\right)=k_{p}\times e\left(t\right)+k_{i}\times \int_{0}^{t}e\left(t\right)dt+k_{d}\times \frac{e\left(t\right)}{dt}$$
where *e*(*t*) is the deviation; *k_p_* is the integral coefficient, and *k_d_* the differential coefficient.

The PID controller was used to realize the closed-loop feedback control of the microgripper, including controlling the tip displacement and gripping force precisely, eliminating the influence of the hysteresis nonlinearity of piezoelectric ceramic, and making the outputs (the tip displacement and gripping force) linear with the input voltages. Figure 8 shows the schematic diagram of the closed-loop control. Where *e* is the displacement or force error; *r* is the given reference displacement or gripping force; *z* is the actual tip displacement or gripping force caused by strain gauges.

In order to verify the actual control effect of the PID controller on the tip displacement, the sinusoidal signal was used as reference input signals in the position tracking experiment. The position tracking curves of the sinusoidal signal with the frequency 0.1 Hz and amplitude 50 μm in the condition of open-loop and closed-loop are shown in Figure 9 and Figure 10, respectively. The results obtained from Figure 9 and Figure 10 were as follows. The control system has obvious hysteresis nonlinearity and the position tracking error was 4.949 μm in the condition of open-loop. In the closed-loop case, the actual output displacement signal and the reference signal are coincident basically, and the significant hysteresis nonlinearity was compensated and reduced to a negligible level. Since the position tracking error was random and obeyed a normal distribution, the standard deviation could be used to show the magnitude of the random error. The standard deviation σ of the position tracking error in the condition of closed-loop was 0.184 μm, which was only 0.37% of the peak-to-peak amplitude of the reference signal.

In order to testify the actual control effect of the PID controller on the grapping force, the sinusoidal signal was used as reference input signals for the position tracking experiment. Figure 11 and Figure 12 show the force tracking curves of the sinusoidal signal with the frequency 0.1 Hz and amplitude 100 mN in the case of open-loop and closed-loop, respectively. It was indicated that the hysteresis nonlinearity was evident in the open-loop condition and the tracking error was 9.71 mN. In the closed-loop case, the hysteresis nonlinearity was controlled availably. The standard deviation σ of the force tracking error was accounted to be 0.225 mN, which was about 0.225% of the maximal amplitude of the reference signal.

It was demonstrated from the above results that due to the hysteresis nonlinearity of the piezoelectric stack actuator, there is a larger error on the tip displacement and gripping force in the open-loop condition. In the closed-loop case, the hysteresis nonlinearity was restrained validly and the closed-loop displacement and gripping force errors obeyed the normal distribution. The standard deviations of the closed-loop displacement errors of the sinusoidal signal do not exceed 0.2 μm and the gripping force errors 0.35 mN. Thus, the PID controller can control the tip displacement and gripping force of the microgripper accurately.

### 3.4. Actual Gripping Test of the Microgripper

A triangular driven voltage with the frequency 0.1 Hz and amplitude 100 V was applied to the piezoelectric stack actuator to manipulate a micro-shaft (Φ 0.9 × 8 mm), and the tip displacement and gripping force of the microgripper changed with time, as shown in Figure 13. It was well known from Figure 13a that the tip displacement increases continuously with the increase of driving voltage and reaches the maximum at 5 s, and then the tip displacement decreases to zero with the decrease of the driving voltage. It was found from Figure 13b that the gripping force is zero before the gripping jaws contact the micro-shaft; the gripping force increases with the driving voltage enhancing after contact and reaches a maximum 64 mN at 5 s, and then decreases with the driving voltage reducing; finally, the gripping jaws and the micro-shaft move away from each other, the gripping force returns to zero.

## 4. Conclusions

A three-stage displacement amplification microgripper based on the compliant mechanism was designed. The theoretical magnification analysis of the microgripper was deduced, and the tip displacement and gripping force were simulated with finite element simulation and, then, the open-loop and closed-loop performances of the microgripper were investigated based on an experimental platform. It was illustrated that the magnification of the microgripper was 16.8. A sinusoidal driving voltage with the amplitude of 100 V was imposed on the piezoelectric stack actuator, and the maximal output displacement of the microgripper was 102.3 μm and the maximal gripping force was 227.7 mN, respectively. It was found that the hysteresis nonlinearity of the piezoelectric stack actuator was eliminated effectively with the closed-loop control. The standard deviation of the displacement closed-loop error was no more than 0.2 μm and the gripping force 0.35 mN. This development of the piezoelectric-driven microgripper can provide some insights to the design and manufacture of micro-electromechanical systems.

## Figures and Tables

**Figure 1 micromachines-11-00121-f001:**
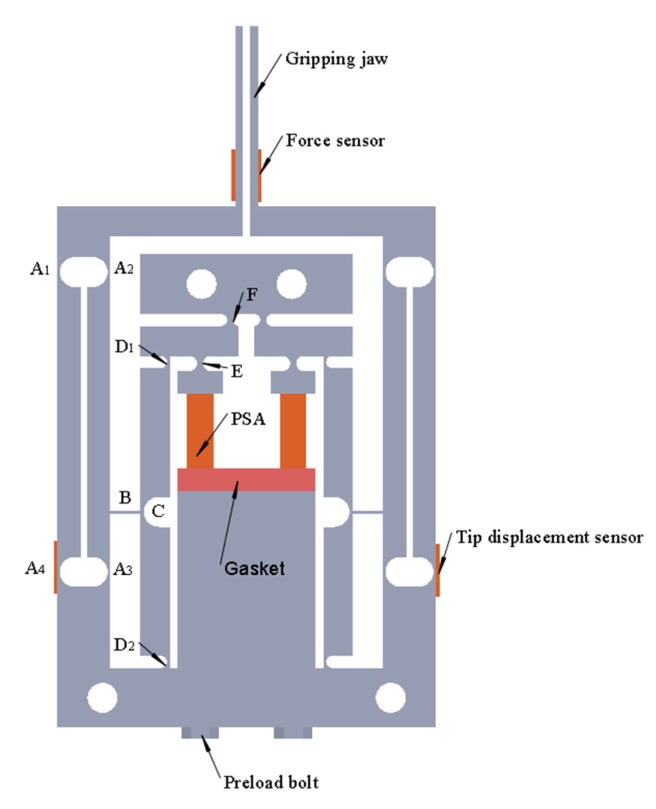
Mechanical structure of the microgripper.

**Figure 2 micromachines-11-00121-f002:**
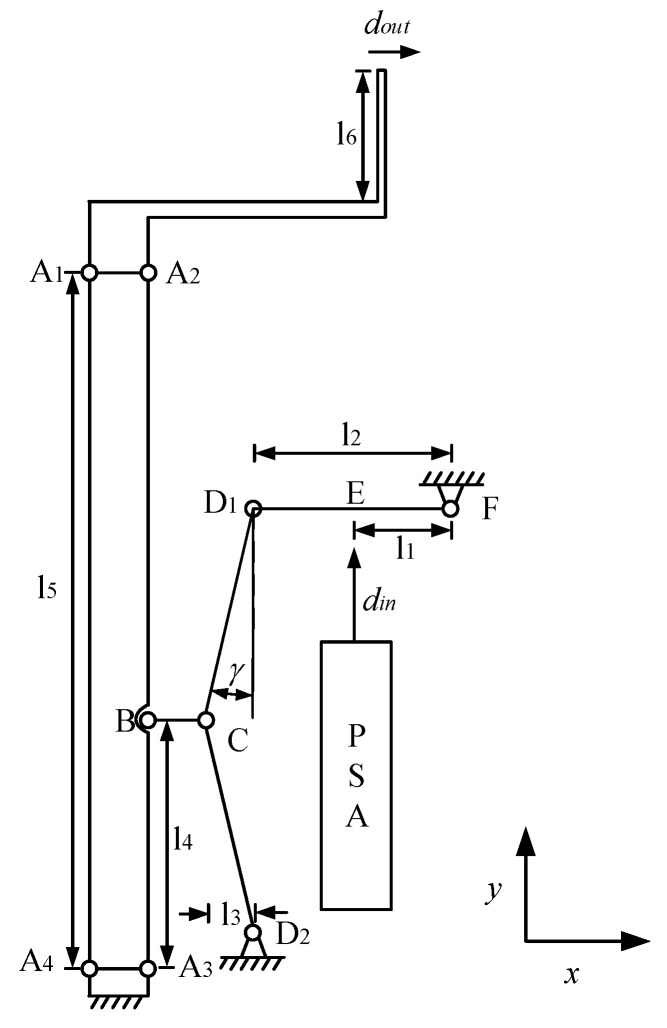
Kinematic diagram of the microgripper.

**Figure 3 micromachines-11-00121-f003:**
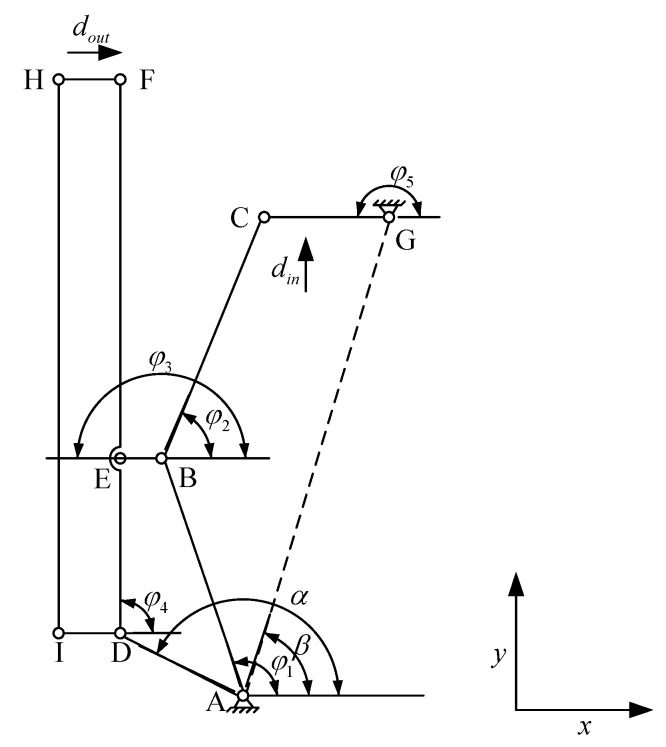
Pseudorigid-body model of the three-stage amplification mechanism.

**Figure 4 micromachines-11-00121-f004:**
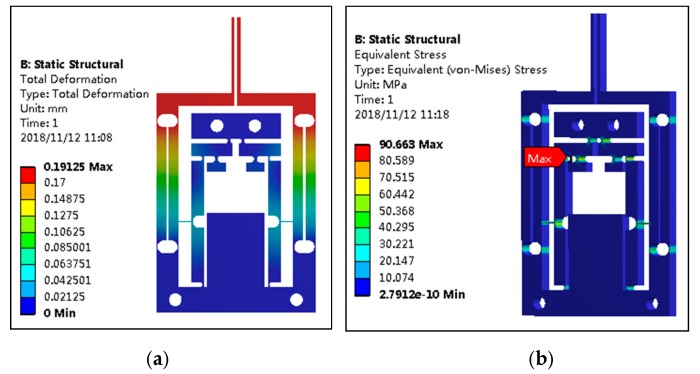
The displacement cloud diagram and stress cloud diagram of the microgripper. (**a**) The displacement cloud diagram; (**b**) the stress cloud diagram.

**Figure 5 micromachines-11-00121-f005:**
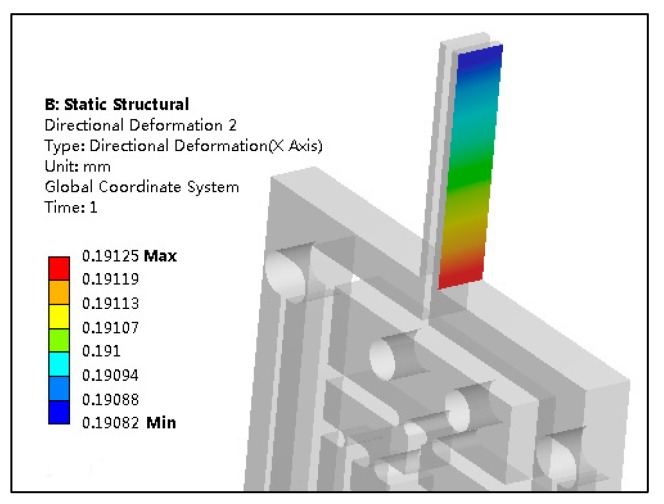
The displacement cloud diagram of the left gripping jaw.

**Figure 6 micromachines-11-00121-f006:**
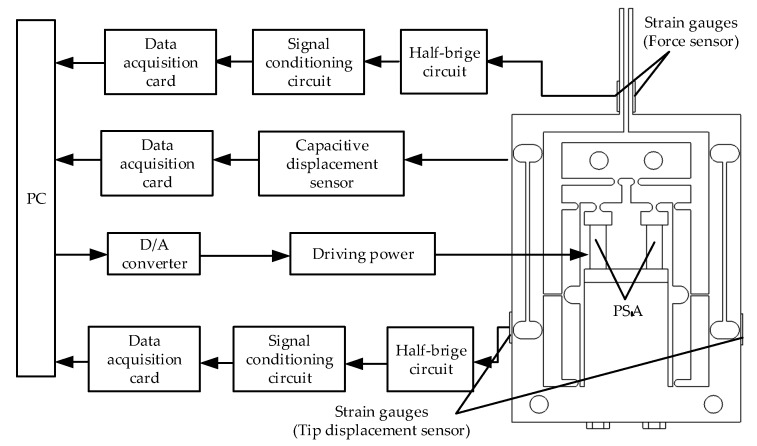
Experimental test system.

**Figure 7 micromachines-11-00121-f007:**
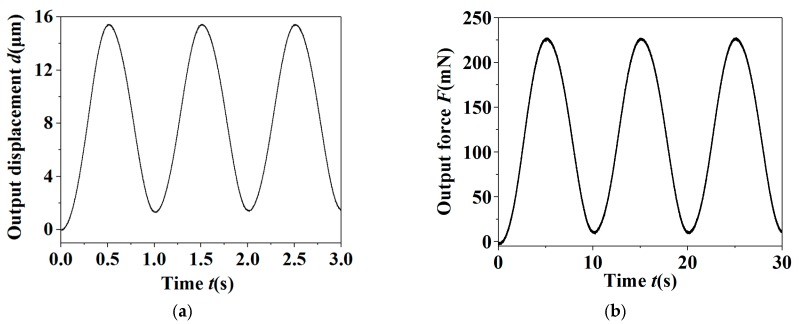
The output displacement and gripping force of the microgripper under a sinusoidal driving voltage. (**a**) The output displacement; (**b**) the output gripping force.

**Figure 8 micromachines-11-00121-f008:**
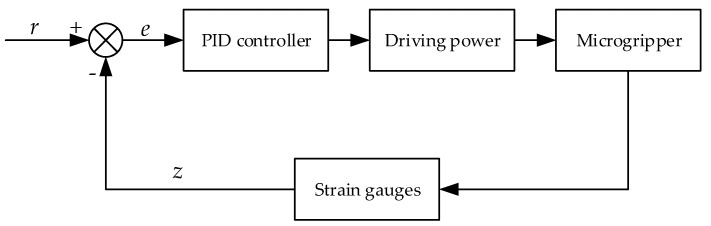
The schematic diagram of the closed-loop control.

**Figure 9 micromachines-11-00121-f009:**
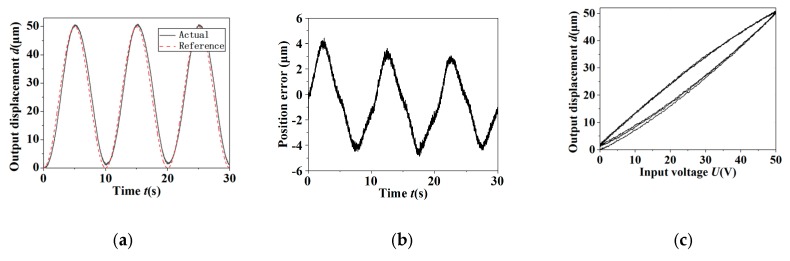
The output results of open-loop control of the sinusoidal signal for the tip displacement. (**a**) The reference signal and output response; (**b**) the error; (**c**) the hysteresis curve.

**Figure 10 micromachines-11-00121-f010:**
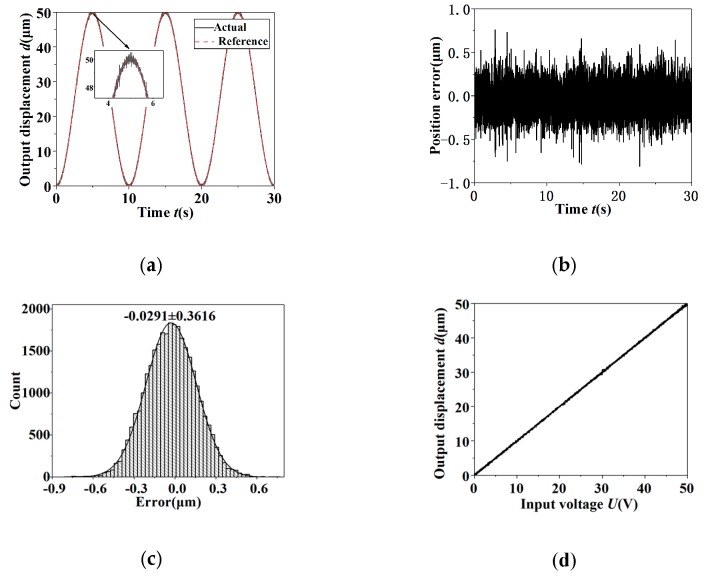
The output results of closed-loop control of the sinusoidal signal for tip displacement. (**a**) The reference signal and output response; (**b**) the error; (**c**) error histogram; (**d**) the hysteresis curve.

**Figure 11 micromachines-11-00121-f011:**
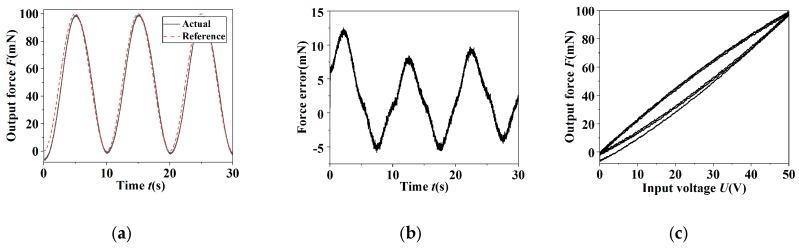
The output results of open-loop control of the sinusoidal signal for the grapping force. (**a**) The reference signal and output response; (**b**) the error; (**c**) the hysteresis curve.

**Figure 12 micromachines-11-00121-f012:**
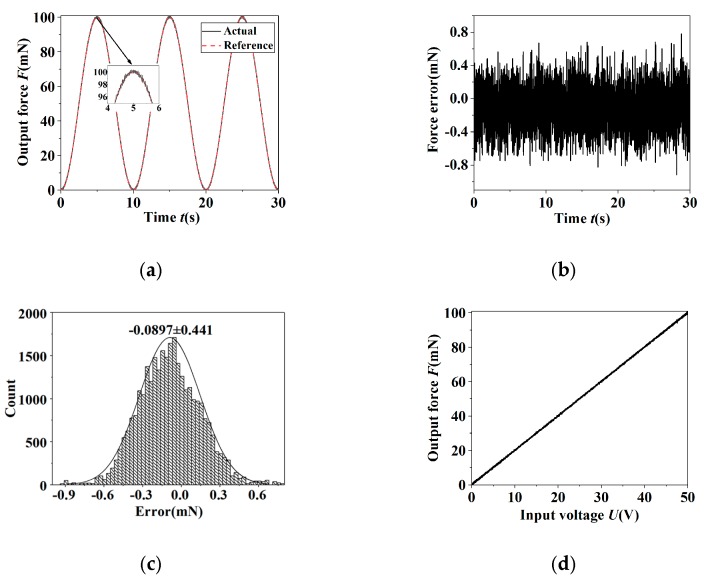
The output results of closed-loop control of the sinusoidal signal for the grapping force. (**a**) The reference signal and output response; (**b**) the error; (**c**) error histogram; (**d**) the hysteresis curve.

**Figure 13 micromachines-11-00121-f013:**
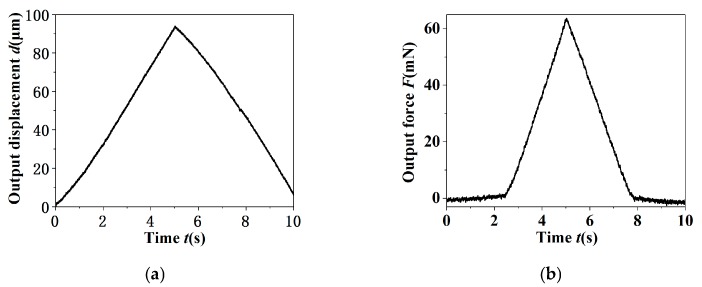
The actual gripping process. (**a**) The tip displacement; (**b**) the gripping force.

**Table 1 micromachines-11-00121-t001:** Parameters of the microgripper.

Parameters	Values	Unit	Parameters	Values	Unit
l_1_	4	mm	r_E_	1	mm
l_2_	8	mm	t_A_	0.3	mm
l_3_	4	mm	t_C_	0.5	mm
l_4_	8	mm	t_D_	0.4	mm
l_5_	40	mm	t_E_, t_F_	0.8	mm
l_6_	24	mm	b	5	mm
γ	12	°	h	1	mm
r_A_, r_C_	2	mm	d	0.4	mm
r_D_, r_F_	0.8	mm	l	4	mm

**Table 2 micromachines-11-00121-t002:** Material properties of the microgripper.

Material	Young’s Modulus (GPa)	Density (kg/m^3^)	Poisson’s Ratio	Yield Limit (MPa)
Aluminum alloy	71.7	2810	0.33	503

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
