# Peer review of "Design and Control of a Piezoelectric-Driven Microgripper Perceiving Displacement and Gripping Force"

_micromachines, 2020, doi:10.3390/mi11020121_

Round 1
Reviewer 1 Report
The authors used a piezoelectric-driven microgripper with 3-stage amplification. The microgripper is able to perceive the tip displacement and gripping force. The key structure parameters were determined by finite element optimization and its theoretical amplification ratio. PID controller is used for tracking experiment of tip displacement and gripping force. The authors believe the designed microgripper provides reference for realizing high-precision micro assembly tasks.
The paper is well written, however there are some flaws which should be addressed prior to publication.
Specific comments:
The authors are suggested to re-write the experimental setup as it confuses the reader.
Please draw the all components of experimental setup in one figure.
What type of processing is done in the computer as mentioned in subsection (3.1)?
The authors are suggested to include the equations of PID controller according to the usage in paper.
How do the authors justify use of the PID controller?
Author Response
Dear reviewer,
Thank you for your suggestions. All your suggestions are very important. They have important guiding significance for my thesis writing and scientific research work!
Best regards,
Sincerely yours,
Yanru Zhao.

Reviewer 2 Report
The paper describes a quite interesting design for a micro-manipulator. The proposed design would be of interest for the community working in this field, but the authors should clarify a few points to better explain the methodology and the results presented in the manuscript:
line 31: references for vacuum microgripper are missing. It would also be nice to briefly recap pros and cons of each of the proposed approaches, to better justify the design chosen for the microgripper described in the paper. line 48: resistance strain sensors are mentioned here for the first time, with no other details provided. Are these sensors being integrated in the gripper? If so, where? It would be nice to briefly explain why sensing and control was embedded in the platform (in the introduction) and then provide more details about this when describing these elements later on in the text figure 4: not clear what this plot is showing and how this is related to what is shown in figure 3... section 3: explain better the setup...where were strain gauges positioned? how was the force calculated? what object was used to evaluate the grasping force? etc... section 3.3: elaborate a bit more on the control scheme used to implement PI control: what are the inputs to the controller? What are the outputs? what are the controller gains? how was the controller tuned? line 151: the Gaussian nature of the noise was only postulated, not demonstrated. Indeed, it would be nice if the authors elaborate a bit on the claim of Gaussian distribution made on line 145 first...has this hypothesis been tested experimentally (e.g. by plotting the histogram of the noise on the error signal)?
Also, quality of English needs to be improved to meet the required quality standard, I would suggest the authors to carefully go through the manuscript to polish the text (ideally, asking a native English speaker to help in the process, if possible). Some suggestions:
line 17: "the control of closed loop" --> "closed-loop control" end of line 17: what does "it" refer to? should it read "this work"? line 23: remove the "the" in front of "microassembly" line 29-31: all the "microgripper" should read "microgrippers" end of line 34: "the" should read "a" equation (5): define "omega"...are omega_1, omega_2 etc simply the time derivatives of phi_1, phi_2 etc? line 137: the comma after "displacement" should be a full stop figure 9: the caption should be on the same page of the figure end of line 179: "experiment" should read "experimental" line 180: not sure what the authors mean with "illuminated"...
Author Response

(The authors gave the same response as above.)

Round 2
Reviewer 1 Report
I would like to accept the paper in present form.